# A Ten-Year Real-Life Experience with Pazopanib in Uterine Leyomiosarcoma in Two High-Specialized Centers in Italy: Effectiveness and Safety

**DOI:** 10.3390/cancers16010192

**Published:** 2023-12-30

**Authors:** Mara Mantiero, Marta Bini, Maggie Polignano, Luca Porcu, Roberta Sanfilippo, Chiara Fabbroni, Gabriella Parma, Mariateresa Lapresa, Carmelo Calidona, Cecilia Silvestri, Andrea Franza, Francesco Raspagliesi, Nicoletta Colombo, Monika Ducceschi

**Affiliations:** 1Department of Gynecologic Oncology, National Cancer Institute of Milan, Via Venezian 1, 20133 Milan, Italy; maggie.polignano@istitutotumori.mi.it (M.P.); francesco.raspagliesi@istitutotumori.mi.it (F.R.); monika.ducceschi@istitutotumori.mi.it (M.D.); 2Department of Medical Oncology, National Cancer Institute of Milan, Via Venezian 1, 20133 Milan, Italy; marta.bini@istitutotumori.mi.it (M.B.); roberta.sanfilippo@istitutotumori.mi.it (R.S.); chiara.fabbroni@istitutotumori.mi.it (C.F.); cecilia.silvestri@istitutotumori.mi.it (C.S.); andrea.franza@istitutotumori.mi.it (A.F.); 3Cancer Research UK Cambridge Institute, University of Cambridge, Li Ka Shing Centre, Robinson Way, Cambridge CB2 0RE, UK; luca.porcu@gmail.com; 4Gynecologic Oncology Division, European Institute of Oncology, Via Giuseppe Ripamonti, 435, 20141 Milan, Italy; gabriella.parma@ieo.it (G.P.); maria.lapresa@ieo.it (M.L.); carmelo.calidona@gmail.com (C.C.); nicoletta.colombo@ieo.it (N.C.); 5Department of Obstetrics and Gynecology, AOUI Verona, University of Verona, Piazzale Aristide Stefani 1, 37126 Verona, Italy; 6University of Milan-Bicocca, Piazza dell’Ateneo Nuovo 1, 20126 Milan, Italy

**Keywords:** uterine leyomiosarcoma, pazopanib, effectiveness, safety, survival

## Abstract

**Simple Summary:**

Pazopanib is an oral drug for metastatic pretreated uterine leiomyosarcoma that received approval in 2012, but poor data have been reported on its activity in real life since the disease is very rare. Uterine leiomyosarcoma has a poor objective response rate to other agents. We assessed the effectiveness and safety of pazopanib in everyday clinical practice, showing its activity and tolerability in patients.

**Abstract:**

Background: Uterine leiomyosarcoma (uLMS) is characterized by aggressive behavior associated with a high risk of relapse and mortality. Several therapeutic agents have been employed in the treatment of metastatic disease, with a poor objective response rate. Pazopanib, approved in 2012, is a multi-targeted, orally active small molecule that exerts its effects by inhibiting several tyrosine kinases. To date, poor research on real-life data has been conducted. We aimed to assess the effectiveness and safety of the drug in everyday clinical practice. Methods: We present results of multicenter retrospective data on 38 patients with heavily pretreated metastatic uLMS who underwent oral pazopanib during their therapeutic journey. Results: At a median follow-up of 8.6 months, the disease control rate was 55.2%, with 17% partial responses and 15 patients (39.5%) with stable disease. At a median follow-up of 8.6 months, median progression-free survival was 4 months, and median overall survival was 19.8 months. The most common grade 3 adverse events (AEs) drug-related were hepatic toxicities, diarrhea, hypertension, nausea, and vomiting (all of them with an incidence of 5% considering the whole study cohort). No grade 4 AEs occurred. Conclusions: Pazopanib in everyday clinical practice is safe and shows a good disease control rate with prolonged survival.

## 1. Introduction

Uterine sarcomas represent a heterogeneous group of tumors, comprising 3–7% of all uterine malignant neoplasms and 1% of gynecological malignancies [1,2]. These tumors originate from either the myometrium or the connective tissue of the uterus. Among them, uterine leiomyosarcoma (uLMS) is the most prevalent histological type, accounting for 60% of all uterine sarcomas [3]. Uterine sarcomas have significantly different behaviors and different responses to drugs with respect to endometrial carcinomas. The more widely used staging system for uterine sarcomas is from the International Federation of Gynecology and Obstetrics (FIGO) [4]. In particular, regarding clinical behavior, patients with uterine sarcomas have more aggressive courses and worse prognoses than patients with endometrial carcinomas. Of note, their more aggressive clinical behavior does not depend on the stage at diagnosis and is often associated with a high risk of relapse and mortality, indicating an unmet medical need [2]. Surgery is the mainstay of treatment in early-stage disease; however, recurrence rates are high. For patients with localized disease confined to the uterus, the risk of relapse stands at approximately 50–70%, highlighting that novel approaches must be implemented [5].

Several therapeutic agents have been employed in the treatment of metastatic uterine sarcoma as both monotherapy and in combined modality, including docetaxel plus gemcitabine, single-agent gemcitabine, doxorubicin, pegylated liposomal doxorubicin, doxorubicin-based regimens, ifosfamide, and trabectedin [6,7,8,9,10]. Unfortunately, these treatments have demonstrated a poor objective response rate (from 9.9% to 36.0%) [6,7,8,9,10]. In addition, a small cohort of uLMS does respond to hormones and sometimes exhibit very durable responses, as in the case of aromatase inhibitors, which has shown a relatively prolonged median survival with an encouraging toxicity profile [11,12].

Pazopanib, an oral multi-targeted drug, comprises small molecules that work actively by inhibiting several tyrosine kinases, as vascular endothelial growth factor receptors (VEGFR-1, -2, -3), platelet-derived growth factor receptors (PDGFR-α and -β), fibroblast growth factor receptors (FGFR-1 and -3), and cytokine receptor (cKIT), which represent its main targets [13].

The randomized controlled phase III PALETTE trial demonstrated the efficacy of pazopanib as a single agent in patients with metastatic soft-tissue sarcomas with previous progression on standard treatment [14]. The study compared two arms: pazopanib versus placebo. In the pazopanib arm, progression-free survival (PFS) was significantly higher than the one in the placebo group, with no difference in terms of overall survival (OS) [14]. Thanks to the result of the PALATTE trial, the Food and Drug Administration (FDA) approved pazopanib in 2012, and the European Medicines Agency (EMA) approved it in 2013. Because of the rarity of the disease and the lack of literature reporting data related to everyday clinical practice [15,16,17], with our retrospective study, we aim to assess the real-life outcome of pazopanib-treated patients with pretreated uLMS in two tertiary specialized oncologic centers in Italy.

## 2. Materials and Methods

A retrospective study on all consecutive uLMS patients who have undergone oral pazopanib between September 2013 and March 2023 was conducted in two high-specialized oncologic centers. Adult uLMS patients aged ≥18 years (with no upper limit of age) having Eastern Cooperative Oncology Group (ECOG) performance status (PS) scores equal to or less than 2, receiving at least one previous line of chemotherapy (CT) for metastatic disease, and having histologically proven uterine high-grade leiomyosarcoma were included. The primary study objective was the effectiveness of pazopanib measured as both response rate and survival outcome. Secondary objectives were the safety and tolerability of pazopanib assessed by capturing all drug-related adverse events (AEs) occurring during treatment.

A further focus was made on the patient cohort that presented an objective response.

The study was conducted in accordance with the Declaration of Helsinki and approved by the Institutional Review Board of IRCCS Istituto Nazionale dei Tumori di Milano (approval id INT 261-23). As for the retrospective design of the study, we received authorization to also analyze data of patients who were deceased or lost to follow-up at the time of data collection. To minimize bias, a shared data dictionary for each variable was provided to all the investigators.

Main clinical characteristics were captured from the patients’ charts. All the information was already registered in patients’ charts per clinical practice. At baseline (i.e., just prior to the first course of pazopanib), we captured the following information: detailed anamnesis, physical examination, basal imaging, echocardiography, age, ECOG PS score, and lab tests, including complete blood counts and serum chemistry panel. For study purposes, the following variables were filled in the study database and subsequently analyzed: histological subtypes, tumor grades, FIGO stage at diagnosis, sites of metastasis, and previous line of treatment/approach for uLMS (primary cytoreductive surgery, use of CT (adjuvant), radiotherapy (RT), previous lines of CT given for metastatic disease, previous surgeries).

Treatment response was evaluated as per clinical practice according to the RECIST, version 1.0. The safety and tolerability of pazopanib were assessed through the evaluation of AEs and serious AEs with the National Cancer Institute Common Terminology Criteria of AEs (v5.0).

Continuous covariates were summarized as median and range; categorical variables were summarized as absolute and percentage frequencies.

Comparisons of groups based on categorical variables were performed using Fisher’s exact test or a chi-squared test when appropriate.

The survival functions were computed by means of the Kaplan–Meier method, with a 95% confidence interval (95%CI).

All tests were two-sided, and a *p*-value of less than 0.05 was considered statistically significant. Statistical analyses were performed with PASS v11 software (NCSS, LLC., Kaysville, UT, USA).

## 3. Results

### 3.1. Patients’ Characteristics

From September 2013 to 2023, 38 women with metastatic uLMS (all high-grade disease) received oral pazopanib in monotherapy. Most patients (95%) had a good ECOG PS score, and the median age was 54.2 years (range of 38–72 years). The most frequent site of metastases was the lung (74%), followed by the pelvis (23.7%) and peritoneum (21.1%), and the median number of previous CT treatments was three (range of two to five) (Table 1). Twenty-six percent of patients had undergone previous RTs.

### 3.2. Treatment, Effectiveness and Outcomes

The median treatment duration was 4.7 months (range of 1.2–21.9 months). At a median follow-up of 8.6 months, the disease control rate was 55.2%, with 17% (n = 6) partial responses (PRs) and 15 patients with (39.5%) stable disease (SD). We also recorded four complicated events that could also be from tumor shrinkage due to response: one pneumothorax, one intestinal perforation, one intestinal occlusion, and one hemoptysis due to pulmonary cavitation. At a median follow-up of 8.6 months, the median PFS was 4 months (95%CI 2.9–6.9), and the median OS was 19.8 months (95%CI of 10.3–25.4) (Figure 1 and Figure 2, respectively).

A total of 82% of patients started pazopanib at 800 mg/die; 42.1% of patients required a dose reduction due to drug-related AEs—4 patients were reduced to 600 mg/die, and 11 patients were reduced to 400 mg/die.

Treatment was continued until disease progression, unacceptable toxic effects, or refusal by the patient. Most patients discontinued pazopanib because of disease progression; two patients (5.2%) definitively interrupted the treatment because of drug-related toxicities (one for anasarca and one for hepatic toxicity), and one patient (2.6%) was still on treatment after 10 months at the time of writing this paper.

Sixty-three patients underwent at least one further line of CT after pazopanib with a median PFS of 3 months—the most frequent CTs were pegylated liposomal doxorubicin (N = 6) and dacarbazine (N = 4).

### 3.3. Safety

A total of 44 drug-related AEs were registered. Seventeen (44.7%) patients had at least one AE of grade 1–2 (mild); 36.8% had one event of grade 3, and 10.5% had two AEs of grade 3 (Table 2). The most common grade 3 drug-related AEs were hepatic toxicities, diarrhea, hypertension, nausea, and vomiting (all of them with an incidence of 5% considering the whole study cohort). No grade 4 AEs or serious AEs occurred.

### 3.4. Patients with Objective Response

Considering the six patients with an objective response to pazopanib, the median age was 52 years (range of 44–69 years) with an ECOG PS score of 0–1. The median number of previous CT treatments was three (range of two to five). No statistically significant difference was found between this subgroup and the cohort of patients who showed progression of uLMS at the first disease assessment after treatment started (Table 3). Fifty percent of them experienced a grade 3 AE (diarrhea, cardiac toxicity with reduced ejection fraction reduction, ocular and skin toxicity), requiring a dose reduction and leading to temporary dose interruption. However, two of them had complicated events that could also be from tumor shrinkage due to response to pazopanib with gastrointestinal perforation and pneumothorax. The median PFS was 6.2 months (95%CI of 4.0 not reached), and the median OS was 10.3 months (95%CI of 7.2 not reached).

## 4. Discussion

Currently, the effective treatment for uLMS is an early resection (mainly complete), and CT is the principal choice in case of unresectable advanced or relapsed cases. To date, clinical trials have shown that no CT regimen (either as a single agent or combination strategy) has superiority over doxorubicin monotherapy as the first-line chemotherapy in these latter cases, mainly in terms of OS. As a second-line treatment and depending on FDA/EMA approval in different countries, pazopanib, trabectedin, and eribulin are used, but their efficacy is not still completely satisfactory, highlighting the unmet medical need and the urgency for the development of novel active agents.

Since FDA and EMA approval, little research has been conducted on the effectiveness and tolerability of oral pazopanib in the uLMS real-life setting, mainly because of the rarity of the disease and the lack of specialized centers.

Our real-life experience with pazopanib includes 38 heavily pretreated patients (median number of previous chemotherapies of three) with poorer conditions than in clinical studies, which have strictly defined inclusion and exclusion criteria. However, our retrospective results confirm the activity of pazopanib in this real-world population—fifty-five percent of patients had a disease control rate, with a median PFS in the overall population of 4 months and an OS of 19.8 months.

The limitations of our research are the retrospective nature of the study, its non-randomized design, and the absence of a control group. The strengths of the research were the homogeneity of the population and the sample size compared to the literature published in a real-life setting.

However, our results confirm those achieved in the phase III PALETTE trial (median PFS of 4.6 months and median OS of 12.5 months) with selected patients and subsequent analyses [14,18,19,20]. To note, our median OS was slightly higher than that of the PALETTE trial. On the other hand, in the PALETTE study, all soft-tissue sarcomas (metastatic non-adipocytic soft-tissue sarcoma) were included, not just those of uterine origin, like in our research. This can justify a slightly different performance in OS [14,18,19,20].

To date, only three retrospective studies have been conducted in a real-world setting [15,16,17] (Table 4). The largest study refers to the pazopanib compassionate program in which 40 uLMS patients were analyzed. Besides that analysis, our study is the largest one conducted in a real-life setting on oral pazopanib used in advanced stage/recurrent uLMS. The disease control rate ranges from 46% to 75%, with a mean PFS of 4 months (range of 2.8–5.8) and a mean OS of 13.8 (range of 6.5–20.0) (Table 4).

We also recorded four complicated responses; thus, considering this possible issue in the context of a rare pathology, it is always recommended to refer patients to high-level specialized centers, as in our case.

Regarding safety, grade 3 toxicities were rare (no grade 4 AEs occurred). Each grade 3 toxicity had an incidence of less than 5%, considering the whole study sample: the drug resulted overall as safe for the patients and easily manageable for clinicians. To date, the safety profile in this study is in line with that reported in the drug characteristics of the product (e.g., hepatic toxicity), with no suspected adverse reactions to report. Patients in PR presented a high rate of grade 3 AEs (50%) with two complicated events that could also be from tumor shrinkage due to responses (intestinal perforation and pneumothorax). Nevertheless, the grade and type of AEs were consistent with the known safety profile of pazopanib [14,15,16,17,18,19,20]. In addition, our rate of discontinuation rate due to AEs was 5.2%, lower than that reported in the literature, i.e., about 13% [21,22].

Although new drugs have been studied/explored in recent years for the treatment of advanced and recurrent uLMS, we currently do not have new molecules available to offer patients except the ones within a clinical trial context or under compassionate/off-label use. Anti-angiogenetic approaches have been explored in patients with uLMS; however, both sunitinib and bevacizumab failed to show sufficient activity in this setting [23,24,25]. Other target therapies, namely nivolumab and pembrolizumab, showed promising results, but to date, they have not yet been approved by EMA for uLMS [26,27].

Discovering and implementing new drug therapies or sequential treatment approaches for unresectable advanced or relapsed uLMS are nowadays an urgent clinical challenge. Although the therapeutic efficacy of several antitumor agents, molecular-targeted drugs, and hormonal therapy have been investigated, there is no strong evidence of a satisfying tumor-reducing or control effect for uLMS patients with a previous history of CT, especially in heavily pretreated cases. Clinical trials incorporating biomarker-based patient selection as inclusion criteria for uLMS patients have not yet been conceived, and we claimed for future studies based on pharmacogenomics. This can help this particular setting with patients characterized by poor prognoses and conditioned by the rarity of the disease.

The research reports no substantial differences from a clinical point of view between patients who had an objective response and those who did not. Thus, to better understand any prognostic factors for response, further studies should be conceived from a molecular point of view, as biomarkers predictive of both response and resistance to pazopanib have still not been identified [28,29]. In fact, prospective studies on pazopanib and subsequent subgroup analyses on different types of soft-tissue sarcomas have failed to pick up (baseline) clinical or pathological features that can improve the use of this drug in terms of both response and survival [28,29].

## 5. Conclusions

Pazopanib in everyday clinical practice showed a good disease control rate with related prolonged survival outcomes. In addition, toxicities were mild and easily manageable, indicating pazopanib as a useful therapeutic tool for heavily pretreated uLMS patients and eligible for further treatment. Physicians should consider this agent in the therapeutic pathway of advanced/recurrent uLMS.

## Figures and Tables

**Figure 1 cancers-16-00192-f001:**
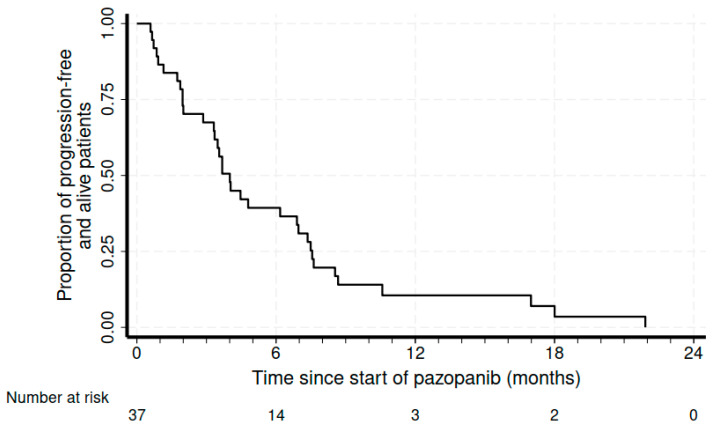
Kaplan–Meier curve for progression-free survival.

**Figure 2 cancers-16-00192-f002:**
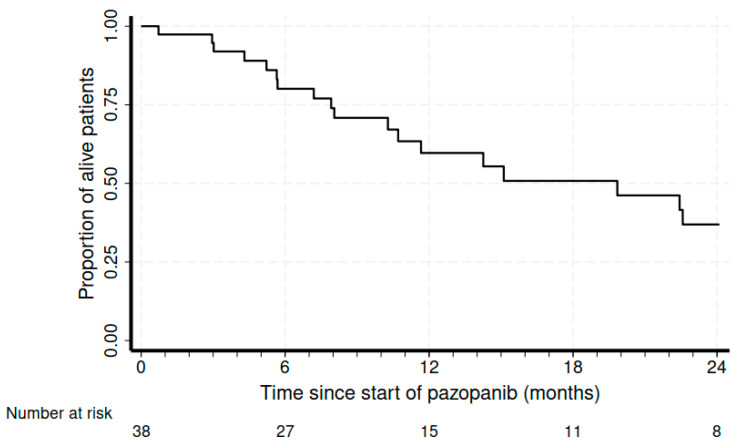
Kaplan–Meier curves for overall survival.

**Table 1 cancers-16-00192-t001:** Patients’ characteristics at baseline.

Characteristics	
FIGO stage at diagnosis, n (%) Stage I Stage II–III Stage IV	13 (34,2) 6 (15.8) 19 (50)
Age at baseline, years median (range)	54.2 (38–72)
ECOG PS at baseline, n (%) 0 1 2	17 (45) 19 (50) 2 (5)
Metastatic sites at baseline, n (%) Lung Liver Bone Peritoneum Lymph nodes Pelvis Soft tissue	28 (73.7) 7 (18.4) 7 (18.4) 8 (21.1) 1 (2.6) 9 (23.7) 4 (10.5)
Previous surgeries, median (range) n (%) 1 2 >3	2 (1–6) 22 (57.9) 11(29.0) 5 (13.1)
Previous lines of chemotherapy median (range) n (%) 2 3 ≥4	3 (2–5) 5 (13.1) 24(63.1) 9 (23.8)
Neoadjuvant treatment, n (%)	12 (31.5)
Type of previous chemotherapies, n (%) Adriamicine Trabectidine Gemcitabine Docetaxel Pegylated liposomal doxorubicin Dacarbazine Ifosfamide Eribulin Epirubicin	34 (89.5) 35 (92.1) 37 (97.4) 19 (50.0) 1 (2.6) 31 (81.6) 8 (21.1) 2 (5.3) 3 (7.9)
Previous radiotherapy, n (%)	10 (26.3%)

ECOG PS, Eastern Cooperative Oncology Group performance status score.

**Table 2 cancers-16-00192-t002:** Pazopanib-related toxicities.

Toxicities, n (%)	Total	Grade 1–2	Grade 3
Neutropenia	2 (4.5)	1 (2.3)	1 (2.3)
Fatigue	8 (18.1)	7 (15.9)	1 (2.3)
Cardiac toxicity	1 (2.3)	0	1 (2.3)
Hepatic toxicity	4 (9.0)	2 (4.5)	2 (4.5)
Nausea	3 (6.8)	1 (2.3)	2 (4.5)
HFS	2 (4.5)	2 (4.5)	0
Vomiting	3 (6.8)	1 (2.3)	2 (4.5)
Diarrhea	4 (9.0)	2 (4.5)	2 (4.5)
Skin	1 (2.3)	0	1 (2.3)
Blurred vision	1 (2.3)	0	1 (2.3)
Anasarca	1 (2.3)	0	1 (2.3)
Blood hypertension	6 (13.6)	4 (9.0)	2 (4.5)
Fever	1 (2.3)	0	1 (2.3)
Interstitial pneumonia	1 (2.3)	0	1 (2.3)
Loss of appetite	3 (6.8)	3 (6.8)	0
Epistaxis	1 (2.3)	1 (2.3)	0
Arthralgia	2 (4.5)	2 (4.5)	0

HFS, hand-foot syndrome.

**Table 3 cancers-16-00192-t003:** Characteristics of patients with objective response versus patients without response with pazopanib.

	Patients with Objective Response (n = 6)	Patients with No Response (n = 17)
Best response	PR	PD
Age at pazopanib, years, median (range)	52 (44–69)	51 (38–72)
ECOG PS, n (%) 0–1 2	6 (100) 0 (0.0)	14 (82.3) 3 (17.7)
Previous CT, n (range)	3 (2–5)	3 (3–5)
Complicated events, n (details)	2 (gastrointestinal perforation; pneumothorax)	1 (hemoptysis)

CT, chemotherapy; ECOG PS, Eastern Cooperative Oncology Group performance status score; PD, progression of disease; PR, partial response.

**Table 4 cancers-16-00192-t004:** Pazopanib real-life experience studies (all retrospective).

Study	Description	Type of Sarcoma	N (All)	N uLMS	mPFS, Months	mOS, Months	DCR, %
Sunar et al., 2019 [15]	Multicenter	Metastatic uterine sarcoma	28	25	5.2	11.4	75.0
Gelderblom et al., 2017 [17]	Multicenter	Advanced soft-tissue sarcoma	211	40	3.0	11.1	46.0
Kim et al., 2018 [16]	Monocenter	Soft-tissue sarcomas	35	27	5.8	20.0	60.0
Present study	Multicenter	uLMS	38	38	4.0	19.8	55.2

DCR, disease control rate; OS, overall survival; PFS, progression-free survival; uLMS, uterine leiomyosarcoma.

## Data Availability

Data for quality control can be made available upon request to the corresponding author.

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
