# Peer review of "A Ten-Year Real-Life Experience with Pazopanib in Uterine Leyomiosarcoma in Two High-Specialized Centers in Italy: Effectiveness and Safety"

_cancers, 2023, doi:10.3390/cancers16010192_

Round 1
Reviewer 1 Report
Comments and Suggestions for Authors
The reviewer recognizes the scientific soundness of a well-structured and pleasant manuscript. The study holds significant importance for the management of uLMS.
However, the authors have not acknowledged the limitations of the study, including its retrospective nature, non-randomized design, and absence of a control group. Hence, the reviewer recommends discussing the strengths of the study in a separate paragraph.
Additionally, it would be advantageous to incorporate a distinct "Perspectives" section, offering insights for future research based on the authors' expertise or opinion.
Could the authors provide an explanation for the (not so) slightly higher overall survival compared to the PALETTE trial? Is this difference significant given the sample size?
The agreement number should be added (l87).
Suggestion: the referee has found recent literature on ULMS and pazopanib, and absent of the reference list. The authors are invited to revise/update their manuscript accordingly.
Comments on the Quality of English Language
The English language requires minor improvement in terms of grammar and spelling.
table3 hemoftoe, remove "range" in right column
l125 emoftoe
l129 #/day
l131 progres¬sion
l194 easily manageable
l212 different types
l213in terms of
Author Response
Manuscript ID: cancers-2760299
Type of manuscript: Article
Title: A Ten-years Real-life Experience with Pazopanib in Uterine
Leyomiosarcoma in Two High-specialized Centers in Italy: Effectiveness and
Safety
Reviewer#1
Dear Professor,
Many thanks for your e-mail regarding this manuscript. You have raised some interest points, and we have now revised the manuscript, in order to incorporate your comments.
Newly added or revised text is indicated in a point-by-point fashion in this letter and with the tracked-changes modality on the original manuscript. In addition, some typos found during the revision were corrected.
Thanks for giving us the opportunity to submit a revised version of the manuscript to Cancers.
Yours sincerely,
Mara Mantiero, MD
Department of Gynecologic Oncology, National Cancer Institute of Milan,
via Venezian 1, 20133 Milan, Italy
Phone: +3902 23903272
mara.mantiero@istitutotumori.mi.it
ORCID ID: 0000-0002-1016-4995
Reviewer Comments:
The reviewer recognizes the scientific soundness of a well-structured and pleasant manuscript. The study holds significant importance for the management of uLMS.
Many thanks for your appreciation.
However, the authors have not acknowledged the limitations of the study, including its retrospective nature, non-randomized design, and absence of a control group. Hence, the reviewer recommends discussing the strengths of the study in a separate paragraph.
In total agreement with you, we acknowledged the limitations of the study, please see lines #190/191 and strengths in lines #198/199.
Additionally, it would be advantageous to incorporate a distinct "Perspectives" section, offering insights for future research based on the authors' expertise or opinion.
Perspectives for future research are reported in lines #224-230.
Could the authors provide an explanation for the (not so) slightly higher overall survival compared to the PALETTE trial? Is this difference significant given the sample size?
in the PALETTE study all soft tissue sarcomas (metastatic non-adipocytic soft-tissue sarcoma) were included, not just those of uterine origin. This could justify a slightly different performance in the overall survival. A sentence was added to comment.
The agreement number should be added (l87).
ID agreement was added.
Suggestion: the referee has found recent literature on ULMS and pazopanib, and absent of the reference list. The authors are invited to revise/update their manuscript accordingly.
Two new references were added as per referee’s suggestion. Please see new References #11 and #12 (Asano et al 2022 and Maccaroni et al 2022). The references list was shifted accordingly.
Comments on the Quality of English Language
The English language requires minor improvement in terms of grammar and spelling.
table3 hemoftoe, remove "range" in right column.
Done, not range but details.
l125 emoftoe
l129 #/day
l131 progression
l194 easily manageable
l212 different types
l213in terms of
Grammar and spelling were improved

Reviewer 2 Report
Comments and Suggestions for Authors
An interesting and important report of real world experience . This is useful additional support for an area of great unmet need.
The results concord with other experiences, it looks like a subset do better and get a modest response whereas there is a cohort who clearly progress quickly and where the drug is of no benefit. Clearly evidence of early progression should lead to cessation.
The authors do not seem to be able to identify the predictive factors in the responders. It would be interesting to try and retrieve their original tissue and look for molecular testing (another paper perhaps if retrievable)
Page 2 lines 49-53, it should be noted that a small cohort of uLMS do respond to hormones and sometimes very durable responses.
Page 4 lines 124, i dont think the term complicated response is appropriate. A response has other connotations, suggest say complicated reactions or events??
Line 125 What is emoftoe?? New word for me! never heard of it after 40 years experience in Gyn Oncology
In summary just needs minor revision
Page 7 line 189 , same comment regarding "complicated response"
Comments on the Quality of English LanguageComments as above
Author Response
Manuscript ID: cancers-2760299
Type of manuscript: Article
Title: A Ten-years Real-life Experience with Pazopanib in Uterine
Leyomiosarcoma in Two High-specialized Centers in Italy: Effectiveness and
Safety
Reviewer#2
Dear Professor,
Many thanks for your e-mail regarding this manuscript. You have raised some interest points, and we have now revised the manuscript, in order to incorporate your comments.
Newly added or revised text is indicated in a point-by-point fashion in this letter and with the tracked-changes modality on the original manuscript. In addition, some typos found during the revision were corrected.
Thanks for giving us the opportunity to submit a revised version of the manuscript to Cancers.
Yours sincerely,
Mara Mantiero, MD
Department of Gynecologic Oncology, National Cancer Institute of Milan,
via Venezian 1, 20133 Milan, Italy
Phone: +3902 23903272
mara.mantiero@istitutotumori.mi.it
ORCID ID: 0000-0002-1016-4995
Reviewer Comments:
An interesting and important report of real world experience. This is useful additional support for an area of great unmet need.
The results concord with other experiences, it looks like a subset do better and get a modest response whereas there is a cohort who clearly progress quickly and where the drug is of no benefit. Clearly evidence of early progression should lead to cessation.
Many thanks for your appreciation.
The authors do not seem to be able to identify the predictive factors in the responders. It would be interesting to try and retrieve their original tissue and look for molecular testing (another paper perhaps if retrievable)
Following your right suggestion, it will be considered for further research. Unfortunately, we not have all original tissues. Thus, subsequent step will be a prospective study with a particular focus on molecular testing. To note, this was acknowledged in the Discussion section.
Page 2 lines 49-53, it should be noted that a small cohort of uLMS do respond to hormones and sometimes very durable responses.
A sentence was added.
Page 4 lines 124, i dont think the term complicated response is appropriate. A response has other connotations, suggest say complicated reactions or events??
Thank you for your right observation. The sentence was reworded in “complicated events that could be also from tumor shrinkage due to response”.
Line 125 What is emoftoe?? New word for me! never heard of it after 40 years experience in Gyn Oncology
In total agreement with you we changed in “hemoptysis”.
In summary just needs minor revision
Page 7 line 189 , same comment regarding "complicated response"
Thank you for your right observation. The sentence was reworded in “complicated events that could be also from tumor shrinkage due to response”.
Comments on the Quality of English Language
Comments as above
